# What’s behind the Dashboard? Intervention Mapping of a Mobility Outcomes Monitoring System for Rehabilitation

**DOI:** 10.3390/ijerph192013303

**Published:** 2022-10-15

**Authors:** Claudine Auger, Cassioppée Guay, Alex Pysklywec, Nathalie Bier, Louise Demers, William C. Miller, Dominique Gélinas-Bronsard, Sara Ahmed

**Affiliations:** 1Faculty of Medicine, School of Rehabilitation, Université de Montréal, Montréal, QC H3C 3J7, Canada; 2Centre for Interdisciplinary Research in Rehabilitation of Greater Montreal (CRIR), Montréal, QC H3S 1M9, Canada; 3Centre de Recherche de l’Institut Universitaire de Gériatrie de Montréal, Montreal, QC H3W 1W5, Canada; 4Department of Occupational Science and Occupational Therapy, University of British Columbia, Vancouver, BC V6T 1Z4, Canada; 5School of Physical and Occupational Therapy, McGill University, Montréal, QC H3A 0G4, Canada

**Keywords:** internet-based interventions, intervention mapping, self-management devices, remote assistance, remote monitoring

## Abstract

Training and follow-up for older adults who received new assistive technology can improve device use adoption and function, but there is a lack of systematic and coordinated services. To address this gap, the Internet-based MOvIT+™ was designed to provide remote monitoring and support for assistive technology users and their caregivers. This paper presents the intervention mapping approach that was used. In step 1, we established a project governance structure and a logic model emerged from interviews with stakeholders and a systematic review of literature. In step 2, a modified TRIAGE consensus process led to the prioritization of thirty-six intervention components. In step 3, we created use cases for all intervention end users. In step 4, the intervention interface was created through iterative lab testing, and we gathered training resources. In step 5, a two-stage implementation plan was devised with the recruited rehabilitation sites. In step 6, we proposed an evaluation protocol. This detailed account of the development of MOvIT+™ demonstrates how the combined use of an intervention mapping approach and participatory processes with end users can help linking evidence-based, user-centered, and pragmatic reasoning. It makes visible the complexities behind the development of Internet-based interventions, while guiding future program developers.

## 1. Introduction

For older adults with a disability and their caregivers, completing daily or caregiving tasks at home can be challenging [1]. Assistive technology (AT), defined as “assistive products and related systems and services developed for people to maintain or improve functioning and thereby promote well-being” [2], like wheelchairs, augmentative and alternative communication devices or low vision aids, can help users and caregivers with completing these tasks. However, the experience of older adults and their caregivers using AT can be both positive and negative. AT can provide positive psychological effects, such as a sense of increased security [3] and a decrease in caregiver’s burden [4]. Conversely, it can also create problems, such as decreased sense dignity and stigmatization for the AT user or caregiver, or increased difficulties in performance if the AT does not fit adequately the user or caregiver’s needs which could lead to AT abandonment [5,6,7]. Some of these issues can be resolved through post-delivery training and follow-up, leading to better AT use and adoption [8,9,10,11]. The adoption of AT is behavior-based as it involves learning new behaviors around the use of objects and technology and integrating these learnings in one’s daily living [12,13,14]. Thus, adequate AT training would involve behavior change techniques [15]. At present, however, very little is offered to support AT adoption or facilitate access to resources which may help AT users and caregivers adapt their behaviors and use effectively their newly obtained AT. Furthermore, the few existing training programs within rehabilitation centers typically require one-on-one, and face-to-face interactions between the AT user or caregiver and a clinician. Because of geographic barriers and time constraints, these training programs are not necessarily accessible for all AT users and their caregivers. Through the Internet, it would be possible to overcome these obstacles while also providing means of remote monitoring for clinicians when potential issues arise [16,17].

At present, there are no Internet-based interventions centered on AT follow-up that: (1) include the caregiver and the AT user as a part of the target intervention population, and (2) incorporate self-management tools and behavior change techniques to support and train the AT user-caregiver dyad. To address these gaps, we developed an Internet-based intervention called MOvIT+™ (Mobility Outcomes Via Information Technology; www.movitplus.com accessed on 10 August 2022). While our initial work embraced AT for various limitations (mobility, vision, communication, …), through the intervention mapping approach we refined the target population to users of mobility aids only: manual and power wheelchairs, wheeled bases, scooters, rollator walkers, walkers and quadripod canes. As such, the Internet-based intervention aimed to address abovementioned gaps by providing non-emergency remote monitoring, support, and training resources to facilitate behavior change in older adult AT users and their caregivers, and to mitigate negative outcomes that can be associated with mobility AT use [18,19,20,21]. With MOvIT+™, remote monitoring is done via self-report questionnaires completed by the AT user or caregiver through customized data collection methods (online, SMS or interactive voice-response system), clinician report (clinical portal) and internet of things (sensors attached to a device post-delivery). Based on the data collected through remote monitoring, the algorithm determines if the detected problem can be resolved by an already existing library of resources within MOvIT+™ (e.g., educational videos, information pamphlets, etc.) or if an alert needs to be issued to a rehabilitation center clinician for follow-up.

Internet-based interventions aimed at behavior change, such as MOvIT+™, can be complex and multifaceted. Each component of the intervention should correlate to a desired behavior change outcome [15]. To achieve these behavior changes, several factors must be considered, such as: the components to include in the intervention, the digital literacy of the different end users, the capabilities of the software, and the selection of evidence-based tools and resources for effective online support and follow-up [20,22]. To manage these complexities, the conceptualization and development of MOvIT+™ was guided by an intervention mapping approach [23,24]. Intervention mapping is an iterative multistep process that structures intervention planning, implementation, and evaluation. It provides a framework for conceptualizing, decision-making, documenting, executing and evaluating an intervention program that is grounded in theory and empirics [23].

The objective of the present paper is to document and describe the scientific reasoning, planning, and executing of MOvIT+™ through an intervention mapping approach. A secondary objective is to share a novel decision-making process we conceived to merge scientific evidence with end users’ knowledge and lived experience. The methods used in each of the six steps of the intervention mapping approach are described in detail. The results section presents the specific deliverables for each step. This paper thus supports a deepening of the understanding of the mechanisms and development process underlying Internet-based interventions.

## 2. Materials and Methods

The development of MOvIT+™ employed a user-centered design that was grounded in a six-steps intervention mapping approach [23]. Table 1 describes these steps. While intervention mapping is presented in a linear fashion, steps can be undertaken simultaneously or revisited considering work developed in another step. For MOvIT+™, it was an iterative process, with steps happening concurrently.

### 2.1. Step 1—Logic Model of the Problem

Step 1 consisted primarily of developing a logic model for the intervention that detailed a governance structure and intended intervention outcomes and objectives (further specified in step 2). A needs assessment was also conducted.

#### 2.1.1. Governance Structure

A governance structure was set to distribute responsibilities through committees to guide the progress of the project, and conduct research, knowledge mobilization, and commercialization activities. Academics whose field of research related to aging, rehabilitation, health services, and health technology were invited to partake in the core research team and governance committees. Selected AT users, caregivers and members of community/advocacy organizations supporting older adults and caregivers were also invited to join the governance committees to provide feedback and insights throughout the intervention development and implementation, thus maintaining a continuous user-centered approach. Three rehabilitation centers with AT delivery programs were identified for collaboration in Greater Montreal, Canada. Clinicians and clinical managers from these AT delivery programs were invited to join one or more of the governance committees, as their contributions would help shape and ground the intervention. Finally, partnerships were established early on with one representative from the Ministry of Health, an industrial partner (for the development of software and online components) and a knowledge broker. These partners could also join governance committees in accordance with their respective expertise.

#### 2.1.2. Needs Assessment

A needs assessment was conducted using a phenomenological descriptive exploratory qualitative design [20,25]. Semi-structured interviews with potential intervention users (i.e., AT users, caregivers, and clinicians) focused on their perceptions of (1) intervention users’ needs, including possible intervention components (e.g., educational videos or information pamphlets), and (2) the potential impact of intervention components to improve user and caregiver outcomes. A modified content analysis approach was used to analyze the interviews [26]. We developed a coding guide based on emerging concepts related to needs identification and a risk analysis framework through consensus between researchers [27,28,29]. Identifying the needs and the risks as perceived by potential intervention users allowed for the intervention design to be modified according to the importance of each of the needs and risk factors [29]. Data collected in these interviews was also used to inform the components prioritization process described in step 2. For more information about the needs assessment methods, see [20].

### 2.2. Step 2—Program Outcomes and Objectives; Logic Model of Change

Step 2 focused on the conceptualization of outcomes and objectives, by selecting the components and the behavior change targets for the intervention. The logic model of change was created iteratively with input from the core research team and the governance committees.

Components for the intervention were selected using a modified Technique for Research of Information by Animation of a Group of Experts (TRIAGE) process [30,31]. TRIAGE is a three-stage consensus-based decision-making process (consultation, compilation, consensus) that has been used previously to identify AT components and outcomes through collective agreement [32]. The consultation stage of TRIAGE consists of participants independently generating relevant ideas for the decision at hand. At the second stage, the ideas are compiled together. Finally, at the third stage, individuals come together as a panel to build consensus using a set of five panels on a wall to help them sort through ideas. These five panels are: the “memory” panel, where all the ideas are initially grouped; the “fridge” panel, to deposit ideas that will be revisited later on in the discussion; the “veto” panel, to remove ideas from the discussion altogether, for a lack of appropriate information; the “selection” panel, to identify ideas that are agreed upon for inclusion; and the “garbage”, to identify ideas that are agreed upon for exclusion [31].

Figure 1 explains the modified TRIAGE approach used for the present intervention [33]. The most substantial modification occurred in the first stage, the consultation, where ideas were generated through two full studies: a qualitative study [20] that served as the needs assessment described in step 1 and a systematic review of literature documenting the components and outcomes of Internet-based interventions aimed at behavior change [22]. The systematic review generated a list of intervention components, behavior change techniques, and program outcomes through a narrative synthesis (see [22] for a complete account of the systematic review method). The list of components identified in the systematic review was presented to interviewees who participated in the needs assessment, and they were asked their opinion on each. Additionally, as with a traditional TRIAGE process, interviewees were asked if they had any other ideas for components. The components were subsequently classified and ranked based the on frequency of component recommendation in interviews and if the interviewees were positive or negative about the component [20].

In the second stage of the modified TRIAGE, the data extracted from the systematic review and needs assessment was not simply compiled, but also organized and logically sorted into a visual matrix of possible intervention components. The level of evidence for possible components was evaluated as high, moderate or low using the Grading of Recommendations Assessment, Development and Evaluation criteria (GRADE) [34]. Each component was associated with a targeted behavior change using a behavior change technique taxonomy [15]. The final matrix included components, targeted behavior, expected outcomes on health and well-being (based on the systematic review) and ranking of the component as per the results of the needs assessment.

Finally, in stage three of the modified TRIAGE, core team researchers, the industrial partner and the knowledge broker were invited to partake in the consensus discussion. Using the above-mentioned matrix and the traditional five panels placed on a wall, each component was presented to and discussed by the group to reach consensus. If there was partial agreement, the component was set aside on the “fridge” panel and discussed again later until agreement was reached. Components that were interesting, but quickly identified as not realistic for development in this project were grouped in the “veto” panel. Detailed notes of the TRIAGE process were taken by a research assistant.

### 2.3. Step 3—Program Design

In step 3, program themes, scope, components, and sequence for the intervention were defined.

#### 2.3.1. Program Themes, Components, and Scope

The themes, components and scope of the intervention were determined using the abovementioned systematic review [22] and through a thematic analysis of the notes and meeting material from the modified TRIAGE [33]. The coding of notes and meeting material was conducted by the second author and verified by the first author. Final codes and themes were determined through discussion between the first and second authors.

#### 2.3.2. Program Sequence

To determine the program sequence, we drafted use cases for all the end users involved in the intervention and we mapped out how they would interact with the MOvIT+™ system, from registration to delivery of clinical resources and/or support. Over the course of several meetings with governance committees, multiple use cases were established and for each, the event trigger, the pre-condition for the event, the flow of actions in the scenario, the users involved, and possible exceptions were described. The use cases were circulated between committee members, commented on, and revised.

### 2.4. Step 4—Program Production

Step 4 consisted of the refinement of the intervention delivery process, development of the intervention software itself, and lab testing. This included the development and testing of various monitoring components and of a library of resources. For pragmatic reasons, it is at this stage that the program was circumscribed for users of mobility aids only, as the clinical processes are more standardized and less complex in comparison to other types of AT, such as visual aids or augmented communication devices. Further funding will be sought in the future to continue the development of MOvIT+™ for these types of AT.

#### 2.4.1. Monitoring Components

All possible monitoring questionnaire response combinations were identified. Research assistants tested the online, SMS and interactive voice response system questionnaires by inputting the different response combinations into a live test version of the intervention. Testing verified the correct flow of questions and the appropriate provisioning of resources based on responses through a decision-making algorithm. Additionally, a novel sensor-based monitoring system was designed to track and analyze AT use.

#### 2.4.2. Library of Resources

Training and education resources for AT users and caregivers were also gathered from clinical coordinators and co-researchers. Resources inclusion criteria based on best practices for older adult learning principles were applied to select appropriate resources and identify gaps or areas of need for new resources [35,36,37,38]. New materials for participants were created by research assistants and intervention co-leads in accordance with the same guidelines and best practices.

#### 2.4.3. Lab Tests

Once the software and monitoring components were developed and part of the library of resources was available, lab tests were conducted with dyads of one older AT user and their caregiver to verify the usability, visual design, user engagement, content, and general subjective evaluation of the intervention. These constructs were selected for testing following the framework proposed by Baumel et al. for the assessment of mobile and online interventions [35]. We recruited dyads of caregivers and AT users aged 65 years or older that had obtained in the year before testing at least one mobility AT (e.g., manual wheelchair, powered wheelchair, wheeled base, mobility scooter, rollator walker, walker or quadripod cane). Lab testing was conducted over two sessions in the research laboratory. Dyads completed a selection of use cases, thus testing the intervention process across multiple tasks and stages, from registration to the delivery of online resources for AT use. The dyads gave feedback on the intervention using a think-aloud method. This recorded feedback was actioned upon with the industrial partner continuously throughout lab tests.

### 2.5. Step 5—Program Implementation Plan

Step 5 involved the selection of partner rehabilitation centers as intervention implementation sites and planning the implementation schedule. Potential implementation sites were targeted following input from the governance committees. Once a site agreed to participate, the knowledge broker led an iterative process of creating site-specific implementation toolkits. To do so, meetings and co-designs workshops were planned every six weeks with each site involving intervention champions, the industrial partner, the knowledge broker, the Ministry of Health representative, rehabilitation program managers, clinicians, and other relevant site-specific actors. Empathy mapping techniques were used during these workshops to deepen the understanding of everyone’s needs and limitations during implementation, from clerks responsible of recruitment to clinicians responsible of intervention delivery [39,40]. The codesign workshops were also used to identify how MOvIT+™ could build on existing rehabilitation center capacities and day-to-day clinician routine to facilitate the program implementation (e.g., adapting intervention processes to different rehabilitation centers or identifying what might facilitate or hinder intervention adoption and usage at a site) [41].

### 2.6. Step 6—Evaluation Plan

Step 6 was concerned with creating the evaluation plan of the intervention. Given its wide acceptance and focus of salient implementation outcomes, the Intervention Outcomes Framework was used to conceptualize and select outcomes through a standardized taxonomy [42]. Outcomes of interest and associated measures were first targeted by the knowledge broker and project co-leads and then validated through meetings with the governance committees.

## 3. Results

At present, steps one through five of the intervention mapping process have been fully realized, while step 6 is partially complete, with data collection and analyses underway.

### 3.1. Step 1—Logic Model of the Problem

The logic model outlined a five-year development plan for MOvIT+™. The logic model (Figure 2) detailed the development, execution, and evaluation of the intervention, potential users, possible challenges, and short/long-term outcomes. It was hypothesized that MOvIT+™ will act on modifiable stressors (e.g., safety and security of tasks, time required) and improve self-efficacy and coping strategies (e.g., knowledge and self-efficacy with tasks involving assistive technology) by providing timely remote monitoring, support, and training through a varied range of Internet-based tools to AT users and their caregivers, leading to improvement in outcomes (quality of life, psychological health, physical health, participation).

#### 3.1.1. Governance Structure

The final governance structure consisted of five (5) governing committees. Their composition is detailed in Figure 2. The executive committee met regularly for operational research decisions regarding budgeting and strategic planning. The steering committee provided continuous practical feedback and insights from political, clinical and community perspectives. Members of the steering committee participated in every step of the development of MOvIT+™ to make sure the intervention would correspond to their concrete needs in the field. The knowledge mobilization committee planned for the dissemination of the project results to the research community and the larger public. The commercialization committee developed a business plan and a value proposition to describe the added value of the intervention for various health care programs. Finally, the research management committee oversaw every step of research studies that were conducted around the development, assessment, and implementation of the intervention.

#### 3.1.2. Needs Assessment

A total of thirty interviews were carried out for the needs assessment. The results highlighted two important aspects that informed the intervention development. First, they captured the needs and risks associated with the AT procurement process. The content analysis revealed that potential intervention users described the acquisition of AT as a cyclical process with key moments pre- and post-delivery where needs go unaddressed. Areas of need and risk focused on (1) getting informed and accessing AT safely and securely; (2) ensuring that AT selection was appropriate for the user’s needs and context-specific; (3) training and support post-delivery.

Second, the interviews documented AT users’ perceptions of the intervention and potential implementation barriers. The content analysis showed that the intervention was described as reasonably accessible and could provide reassurance to AT users and their caregivers. The intervention was also perceived to humanize forms of remote communication. Interviewees noted that the intervention could contribute to preventing AT abandonment by providing access to early support and training, especially by mitigating transportation barriers for AT users and caregivers who live in isolated areas. Clinicians and managers perceived that the intervention may facilitate ensuring that intervention objectives were met and securing intervention success. Potential barriers to the intervention implementation identified included lack of political interest or perceived value by decision-makers. Efforts were made to mitigate the impact of identified potential barriers, for example, involving policy and decision makers in the intervention development and execution. For a complete account of the needs assessment results, see [20].

### 3.2. Step 2—Program Outcomes and Objectives; Logic Model of Change

The consensus panel was comprised of five core researchers, the knowledge broker, and the industrial partner (*n* = 7). A total of sixty-six possible intervention components were pre-ranked and organized into a visual matrix. The matrix was then presented for prioritization during two meetings totalizing six hours of discussions. The panel agreed to include thirty-six intervention components and thematically compiled them in three groups: the “monitoring” group, for components that allow clinicians to stay informed of any problems that might arise with the newly delivered AT; the “eExpert” group, for components that enable synchronous or asynchronous remote contact with an expert (i.e., clinician, technical support); and the “resources & training” group, for components that promotes engagement with the intervention and provide resources and online training for AT learning and troubleshooting. Following the analysis of TRIAGE notes and material and based on the logic model, we linked each component to target behaviors, associated determinants of change and expected outcomes in the format of three detailed tables. Each component was also matched with its original ranking from the needs assessment and its supporting evidence from the systematic review [16,43,44,45,46,47,48,49,50,51,52,53,54,55,56,57,58,59,60,61,62,63,64,65,66,67,68,69,70,71,72,73,74,75,76,77,78,79,80,81,82,83,84,85,86,87,88,89,90,91,92,93,94,95,96,97,98,99]. Table 2 provides a summary of all the thirty-six included components. Appendix A present detailed tables with target behaviors, determinants, and outcomes (Appendix A). These detailed tables were subsequently used as the logic model of change for MOvIT+™.

As an example, the component that received the highest ranking was “skill-based videos” because it received the most positive feedback from stakeholders during the needs assessment conducted at step 1 [20]. Its positive effects on behavior change are also documented in two studies [65,77], one of which had high level of evidence as per GRADE criteria [22]. For these reasons and because it was predicted that skill-based videos would enable AT users and their caregivers to better perform tasks with the AT through role modeling, thus reducing the number of problems detected with the AT, the consensus panel retained this component to be developed in MOvIT+™. The main reasons for rejecting a component were either issues with ethics and confidentiality (e.g., home video monitoring and motion sensors) or limited support from literature or stakeholders (e.g., a personal webpage).

### 3.3. Step 3—Program Design

In step 3, a program themes, scope, and sequence for the intervention were defined.

#### 3.3.1. Program Themes, Components, and Scope

The thematic analysis of the notes and recordings of the TRIAGE meetings revealed three themes [33]. First, components were selected by the committee to *reduce potential barriers* to accessing and understanding the intervention. This was achieved by choosing components that were easy-to-use, interactive, and intuitive with clear and obvious benefits for the different users. Second, by providing human support if needed, selected components were seen to encourage *making connections* between clinicians, users, and caregivers by developing their self-efficacy in using the AT and identifying when professional support is needed. Finally, the selected components provided *privacy and protection*, ensuring all users’ personal information was protected [33]. Collectively, the scope of the program was to increase AT users’ and caregivers’ knowledge and confidence while monitoring and reporting on device use. By modifying the way assistive technology is used, it was perceived that MOvIT+™may optimize benefits, avoid harm, and empower users and caregivers to seek help when needed. These themes formed the core values for creating a user-centered Internet-based intervention that may achieve positive behavior change.

#### 3.3.2. Program Sequence

Twenty-eight unique intervention use cases were devised, covering the interaction with MOvIT+™ of all end users (i.e., AT users, caregivers, clinicians, managers, and research staff). The general program sequence and intervention flow (Figure 3) was designed as follows. First, a clinician at a participating rehabilitation center delivers a new AT to a service user. Dyads of AT users and caregivers are then identified as potential participants in MOvIT+™ via lists compiled by administrative staff at the rehabilitation center. Next, AT users or their caregivers confirm enrollment via an email form. At one and three months post-delivery, AT users or their caregivers answer a monitoring questionnaire via their pre-established preferred modality (online, SMS or interactive voice-response system) regarding different aspects of their experience with the new AT. The questionnaire is described here [8]. Based on questionnaire responses, the MOvIT+™ algorithm determine if issues can be addressed with the library of resources. If so, AT users or their caregivers receive corresponding online training resources automatically via email (e.g., instructional videos, information pamphlets, practice exercises). For issues that cannot be addressed automatically online, an email alert is sent to the clinician in charge of the follow-up. The clinician logs into the intervention system (clinical portal) to see questionnaire responses and the identified area(s) of concern. The clinician can then send out additional online resources that they deem appropriate or follow-up directly with the AT user or caregiver.

### 3.4. Step 4—Program Production

Following results from steps 2 and 3 and by adjusting iteratively through lab testing, the final interface and software for the intervention included an automated registration system, multiple monitoring components, a library of resources, an integrated algorithm to facilitate clinical decision-making and dispatching of resources, and a clinical portal (Appendix A).

#### 3.4.1. Monitoring Components

Nine core problems with AT use could be identified with the finalized version of the monitoring questionnaire: non-use, pain/discomfort, restricted participation (e.g., the user can no longer access their bathroom from the wheelchair being too large), skin problem, positioning problem, incident, psychosocial problem (e.g., poor mood as a result of AT use, such as feeling stigma or shame from using the AT and having a visible different mobility) and limited AT skills and knowledge [8]. How answers to the questionnaire are compiled and acted upon in the clinical portal is shown in Appendix A.

The sensor-based monitoring system collects data on the use of power wheelchairs, specifically tracking tilt angle and time spent in the wheelchair (time seated and time tilted). If pre-determined behavior targets are not met, a sound or vibration notification is issued to the AT user as a reminder [21]. The sensor-based monitoring system is also connected to the clinicians’ intervention system. The wheelchair usage data is made available through daily and monthly graphics. The development of this sensor-based monitoring system was prioritized by the steering committee to decrease risk for pressure sores. This sensor-based monitoring system can be used as both a monitoring and educational tool, facilitating AT adoption by outlining to the user the behavior targets and progress with goal attainment.

#### 3.4.2. Library of Resources

Upon publication, the intervention system housed more than a hundred and thirty online resources organized the MOvIT+™ website (Appendix A). Resources included pre-existing online content in both French and English, such as websites, digitized information pamphlets, and videos. New resources were also created specifically for the intervention, including a series of thirty instructional videos in English and French regarding AT use [100], one-page information documents regarding post-delivery procedures for AT, and instructions for the use of the sensors-based monitoring system.

#### 3.4.3. Lab Tests

Five dyads of AT user and their caregiver, using a variety of mobility AT, completed two sessions of testing each in the research laboratory. Bugs and failures within the system were managed continuously as they arose during lab tests. Important adjustments to the registration process and the delivery of online resources were also made. Initially, registration to MOvIT+™ was done through a paper form to be filled by the AT user or their caregiver involving numerous questions, making the process tedious and too long for the staff, which led to changing it to an online form. As for the delivery of online resources, this is done through an email sent to the AT user or their caregiver. At first, a simple hyperlink leading to the selected resource was provided, but through lab testing it was discovered that often the participants did not realize they had to click on the hyperlink. The email was therefore adjusted to include instructions on how to access resources.

### 3.5. Step 5—Program Implementation Plan

With the guidance of the steering committee, participation as an implementation site was sought from home care programs in community clinics, short-term stay geriatric hospital units, hospital specialty clinics (e.g., multiple sclerosis clinic, amyotrophic lateral sclerosis) and AT delivery programs from rehabilitation centers. Only settings involved in AT provision showed interest, resulting in a total of six recruited rehabilitation centers across Quebec, and one private clinic in British Columbia, Canada. The main reason for refusal from the other clinical settings was that the intervention was not directly linked to their service mandates (e.g., while community clinics deal with home adaptations, they do not prescribe mobility AT in Quebec).

The complete implementation toolkit included written guides, visual prompts and reminders, promotional materials, and interactive presentations developed to support every step of implementation, from recruitment of participants to deployment of MOvIT+™. The toolkit was adapted for each site and each actor involved in the implementation process. For example, a visual recruitment map was created for administration staff responsible of generating lists of potential participants and a detailed technological how-to guide was developed for clinicians who would need to understand and use the intervention software.

The iterative implementation planning process with the participating sites and the knowledge broker led to the final implementation plan. It is designed as a two-stage process with a feasibility pilot study in three AT delivery programs followed by a scaleup study in the total six participating rehabilitation centers. For the pilot study, the recruitment target is of forty AT users and their caregivers. For the scaling-up, an intervention period of six months is planned with an estimated sample size of eight hundred participants. Implementation for the scaleup study is set to follow a progressive and overlapping approach with evaluation, meaning that while the intervention is being implemented in one site, it is being evaluated in another further along in the process.

### 3.6. Step 6—Evaluation Plan

The final outcomes selected for the evaluation of the intervention are presented in Table 3 and grouped in two main categories of variables: clinical and implementation. Clinical variables include items such as: the number of problems detected by MOvIT+™, number of problems resolved, and general satisfaction with AT. Implementation variables include items such as: relevance of intervention to stakeholders, intervention acceptability, feasibility, and cost of implementation. Four assessment points were established: the day of AT delivery (T0), one month after delivery of AT (T1), three months after delivery of AT (T2), and six months after delivery of AT (T3). For example, a cost analysis of implementation was added iteratively as it was identified by both the Ministry of Health representant and rehabilitation program managers as a key variable to aid their decision as to whether MOvIT+™ would be implemented permanently after initial testing. Results from the pilot study and the scaleup study will be shared in future publications once fully completed.

## 4. Discussion

The Internet-based intervention MOvIT+™ aims to address a gap in AT post-delivery services by providing remote monitoring, support, and training resources to AT users and their caregivers. This paper described the development of this Internet-based intervention through an intervention mapping approach and highlighted the innovative modifications made to a consensus-based decision-making process (i.e., TRIAGE). Some of the main valuable research observations from this project are discussed here.

The use of the intervention mapping approach is in line with the four principles of the Framework for Accelerated and Systematic Technology-based intervention development and Evaluation Research (FASTER), which are user-engagement, transdisciplinary collaboration, ethical development and implementation and process evaluation and transparency [101]. Specifically, it proved to be the right fit in two areas: (1) managing multiple stakeholders needs and preferences in an ecological manner and (2) documenting the intended mechanisms of change and development rationale.

To our knowledge, MOvIT+™ is the first Internet-based intervention of its kind that is addressing issues arising after delivery of assistive technology for both the user and the caregiver, while also accounting for the realities of involved clinicians, rehabilitation centers and service providers at the government level. Typically, Internet-based interventions for users and their caregivers are delivered individually to only one or the other. Very few Internet-based intervention designs have a dyadic approach [102], despite clear empirically and theoretically demonstrated benefits [103]. Additionally, these Internet-based interventions are often developed and deployed in isolation and outside of current established health care services [22], thus limiting their adoption, integration and promotion by health care practitioners and systems [104]. Taking into account the needs of not only the target population, but also of the actors responsible for implementation and dissemination early in the development process can allow for the intervention to be coherent, eliciting collective engagement and action, as per the recommendations from the normalisation process theory [104]. The development of MOvIT+™ was therefore largely facilitated by the governance structure that was mobilized at every step of the way to provide practical feedback. Furthermore, the intervention mapping approach led to a thorough needs assessment that was both informed by end users and the best available current evidence. The iterative continuous testing along development also helped avoiding the replication of common shortcomings of technology targeting older adults, like unfriendly features to their levels of technology literacy [11]. Early lab tests, for example, helped to pin-point components that needed adaptation (i.e., registration, links in emails) for older adults’ needs before large scale deployment, preventing later difficulties in the implementation and evaluation stages.

The intervention mapping approach also worked as an organizing principle, assisting investigators and governance committees to document innovative connections between what MOvIT+™ does, for whom and what changes are expected. Going through the processes of creating the logic model of change, for example, encouraged stakeholders to clearly communicate their reasoning, both pragmatic and evidence-based, for every need voiced or decision made, thus making explicit the implicit. Furthermore, the thorough documentation of the development, both at theoretical and technological levels, ensures that every decision can be traced back and future users can be informed of the project’s rationale if questions arise. This felt crucial for the years long development process that is necessary for a complex Internet-based intervention like MOvIT+™, preventing loss of important decision-making rationale along the way. Outcomes that will be assessed during the evaluation phase will also be easily connected back to intervention components and the intended behavior change technique through the same logic model of change. This traceability enhances chances of isolating the effects of each individual components. Overall, it corresponds to research recommendations from the field of behavior change theory and Internet-based interventions to be more transparent and detailed about the underlying change mechanisms [105].

### Strengths and Limitations

Further capitalizing on and extending existing approaches to the development of complex interventions, a novel methodological contribution was produced: the modified TRIAGE. The modifications were two-fold; evidence from a systematic review was incorporated to the “consultation” stage, in addition to in-depth interviews with end users, and the results were compiled and pre-ranked into a visual matrix of potential components to be prioritized. It allowed decision-making to be well informed by data triangulation from various methods (i.e., systematic review and qualitative study), yielding a prioritization that is more rigorous and complete [106]. Data triangulation of the like also permitted to overcome one of the main limitations of traditional TRIAGE which is that “quality of the information obtained is directly related to the competence, the representativeness and the credibility of the participants” of the consensus panel [31]. The use of visual ranking in a matrix were also key to process the complexity of the information gathered in the consultation stage into something that was simpler for the consensus panel. The main limitations noted for this modified TRIAGE are the time and resource commitment that went into the consultation and compilation stages. A systematic review combined to a thorough qualitative study might not be within reach of all research projects. The time required to compile the information into a matrix was also substantial and required high-level integration skills. We would however argue that these investments were well worth it, as the final consensus stage itself was short and greatly efficient considering the complexity and quantity of information processed, the overall discussions only taking six hours in total. Besides, another limitation is that the intervention mapping approach does not provide any decision-making tools while prioritization of needs is crucial for any intervention development, as not all needs can be realistically answered. Using a prioritization method such as TRIAGE filled this gap in the intervention mapping approach.

## 5. Conclusions

The development of an Internet-based intervention is a complex and resource intensive process. It is crucial that the planning and implementation process be thorough, rooted simultaneously in the end users’ needs and scientific evidence. By utilizing an intervention mapping approach, the complexities of an Internet-based intervention were manageable. We would encourage future intervention developers to use the modified TRIAGE process for the richness of ideas it generated. Including all important stakeholders would ensure that key decisions about components selection and, by extension, the targeted behavior changes, are informed by end users’ needs and evidence. MOvIT+™ addresses gaps in current AT post-delivery services that can be expanded to other rehabilitation centers, pediatric populations, and countries. It also documents the links between intervention components and behavior changes, building the evidence base for more effective behavior change interventions.

## Figures and Tables

**Figure 1 ijerph-19-13303-f001:**
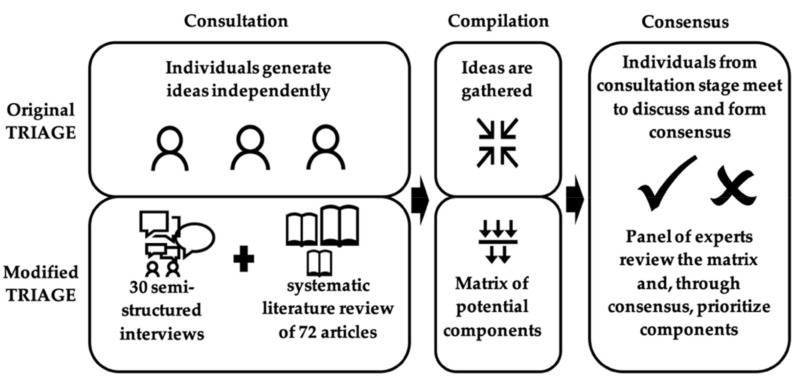
Modified TRIAGE.

**Figure 2 ijerph-19-13303-f002:**
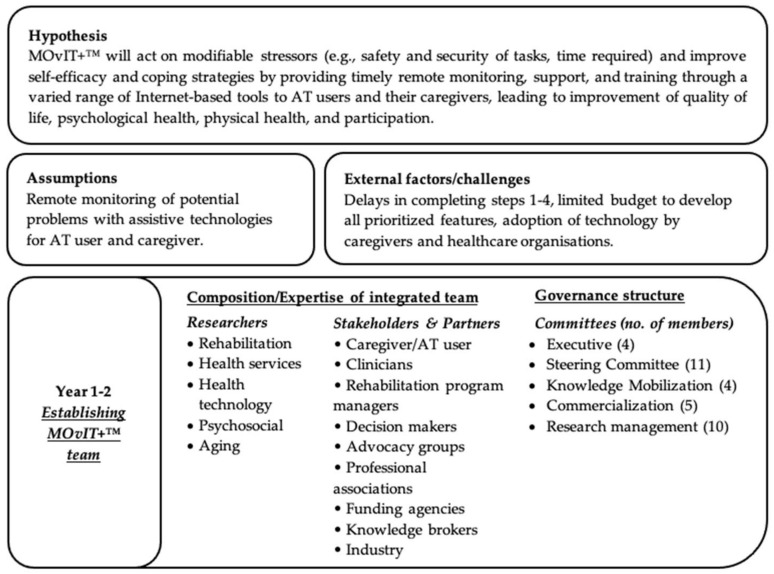
Logic model of the problem.

**Figure 3 ijerph-19-13303-f003:**
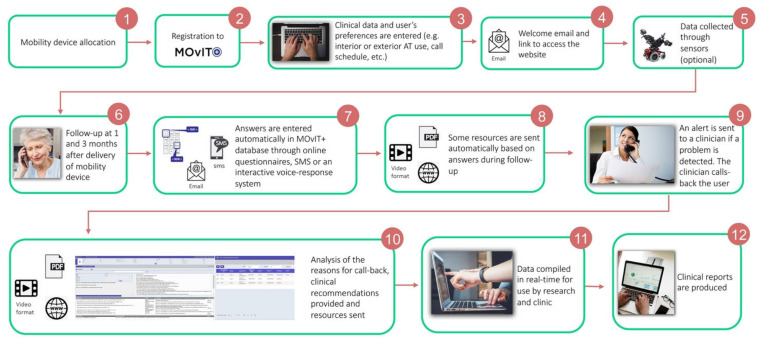
Program sequence.

**Table 1 ijerph-19-13303-t001:** Intervention mapping approach ([23], p.13).

	Step	Description
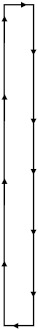	STEP 1: Logic Model of the Problem	Establish and work with a planning groupConduct a needs assessment to create a logic model of problemDescribe the context for the intervention including the population, setting, and communityState program goals
STEP 2: Program Outcomes and Objectives; Logic Model of Change	State expected outcomes for behavior and environmentSpecify objectives for behavioral and environmental outcomesSelect determinants for behavioral and environmental outcomesConstruct matrices of change objectivesCreate a logic model of change
STEP 3: Project Design	Generate program themes, components, scope, and sequenceChoose theory- and evidence-based change methodsSelect or design practical applications to deliver change methods
STEP 4: Program Production	Refine program structure and organizationPrepare plans for program materialsDraft messages, materials, and protocolsPretest, refine, and produce materials
STEP 5: Program Implementation Plan	Identify potential program usersState outcomes and performance objectives for program useConstruct matrices of change objectives for program useDesign implementation interventions
STEP 6: Evaluation Plan	Write effect and process evaluation questionsDevelop indicators and measures for assessmentSpecify the evaluation designComplete the evaluation plan

**Table 2 ijerph-19-13303-t002:** Final list of retained components for MOvIT+.

Component	Rank	Supporting Literature
Monitoring components		
Self-report questionnaires—Online	2	[43,48,54,55,80,87,89,95,99]
Self-report questionnaires—Interactive voice-response system	4	[16]
Self-report questionnaires—SMS	6	None
Sensors recording AT use	7	None
eExpert components		
Clinician support by telephone (synchronous)	3	[47,61,72,87,89]
Clinician support by video conferencing (synchronous)	8	[72,73,90,95]
Clinical alerts	12	[69,95]
Clinician report	14	[48,54,55,80]
Clinician support by email (asynchronous)	16	[46,49,66,73,74,83,84,85,86,87,90,93,94]
Ask an expert section (asynchronous)	26	[48,53,55,79,80,82,89,92,94,95]
Technical support by phone	28	[48,55,73,77,83,84,85,86,90]
Technical support by email	35	[73,77,81,89]
Technical support button	None	[92]
Resources & Training components		
Skill-based videos	1	[65,77]
Decision aids	5	[50,54]
Data base of accessible public places	9	None
Written educational material	11	[43,45,46,48,51,53,55,56,57,63,68,70,71,74,75,77,78,79,80,81,83,84,85,86,88,89,90,91,92,94,96,97,98]
Educational videos	15	[45,54,60,63,68,71,75,78,79,82]
Interactive and thematic classes	17	[51,58,59,66,67,91]
Written caregiver tips	19	[48,55,80,82]
Resource directory	21	[44,48,52,53,55,63,80,82,87,89]
List of relevant web links	29	[47,48,53,55,58,59,60,66,70,73,76,77,80,82,83,84,85,86,92,93,94,96]
Frequently asked questions (FAQ) section	31	[48,53,55,59,80,92]
Videos with testimonials	33	[43,54]
Instant library	35	[48,53,54,55,58,59,77,79,80,87]
Facebook and Twitter links	35	[64,81,82]
Reminders	36	[50,52,62,90]
References library	None	[53,58,59,89]
Search bar	None	[81,93]
Online quizzes/homework	None	[45,56,63,66,70,79,93]

Notes: no ranking for components that were not discussed during needs’ assessment. The mention ‘none’ under ‘supporting literature’, means that this component emerged from the interviews, not from the systematic review.

**Table 3 ijerph-19-13303-t003:** Clinical and implementation outcomes.

Variables	Sub-Variables	Measurement Tool/Source	AT user	Caregiver	Coordinator	Management	Administration Time (min)			
							T0	T1	T2	T3
**Clinical variables**										
Problems	No. detected problems No. resolved issues	Automated questionnaires	X	X				8	8	
Satisfaction with AT	General satisfaction Satisfaction with AT characteristics Satisfaction with online AT services	QUEST	X				10		10	
Satisfaction with participation with AT	Importance of mobility assistance and satisfaction with activities carried out inside and outside. Satisfaction with the positioning. State of the skin.	WhOM	X				10		10	
Impacts AT on caregiver	Field and type of assistance Stressors and burden of AT use Impacts on physical health Impacts on quality of life	CATOM		X			15		15	
Knowledge	What knowledge did you learn with the follow-up intervention?	End of intervention interview	X	X					15	
Behaviors	Tell me about your new ways of doing things or new activities that you have done after seeing the resources provided by MOvIT+™?									
**Implementation variables**										
Relevance	Compatibility Perceived utility/added value of the intervention Sample questions: What are your expectations of the MOvIT+™ project in the short and long term (earnings)? What are the difficulties you expect to encounter with MOvIT+™ (Brake/Loss)	Meeting of managers			X		30			
		Values map				X	30			30
Acceptability	Perceived usability Intent to use Satisfaction with the intervention Sample questions: How do you describe the utility of MOvIT+™? What other problems could be improved by MOvIT+™ (earnings)?	End of intervention interview	X	X					15	
		Feedback at 6 weeks between T1 and T4			X		15			
Feasibility	Rate of intervention adoption by sites and reasons for refusal	Recruitment tracking diagram	N/A				Extracting data from the interface			
	Rate of refusal/exclusion of users and caregivers and characteristics (age, site, AT)									
	Attrition rate and reasons for abandonment	Follow-up chart of abandonments								
	Rate of participation in the intervention	Navigation data on the site MOvIT+™								
Fidelity	Adherence to the intervention protocol	MOvIT+™ navigation data	N/A				Extracting data from the interface			
	Dosage									
Costs	Consultations related to the detected problems	Statistics extracted from the purchase orders and clinical notes found in patients’ records	N/A				N/A			
	AT Repairs									
	Use of health services	Patients’ records summary; convert service time into economic value								
	Technology	Reports of web hosting costs, hardware, technical support resources and TelASK Inc. automated service.								

Abbreviations: AT: Assistive Technology; CATOM: Caregiver Assistive Technology Outcome Measure; QUEST: Quebec User Evaluation of Satisfaction with assistive Technology; WhOM: Wheelchair Outcome Measure; N/A: not applicable. T0: baseline, day of AT delivery; T1: 1 month after AT delivery; T2: 3 months after AT delivery; T3: 6 months after AT deliver.

## Data Availability

Not applicable.

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
