# Peer review of "What’s behind the Dashboard? Intervention Mapping of a Mobility Outcomes Monitoring System for Rehabilitation"

_ijerph, 2022, doi:10.3390/ijerph192013303_

Round 1

Reviewer 1 Report

The authors used adequate methodology and included enough details in different sections with summary tables and figures. This study clearly explained the logic model of the problem and illustrated the program sequence (Figure 4). Minor suggestion to combine qualitative and quantitative measure: Since this is qualitative study, no need to report outcomes of study group. However, I am wondering about Usability metrics, retention, and adherence to intervention. Is there any possibility to report those measures?

Author Response

Table 2 reports the evaluation plan, namely measures of Acceptability (perceived usability) – usability metrics, Feasibility (attrition rate) - retention and Fidelity (adherence) – adherence to intervention. We did not include the actual results in the current paper because it would require 3 additional tables and figures. We mentioned on line 466 that they will be published in a separate future paper. 

Reviewer 2 Report

1. I think Figure 1 is a table, please tabulate the picture.

2. the questionnaire without supporting literature in Table 1 How was it designed?

3. What are the shortcomings of this paper and what are the future directions for improvement that I think should be talked about.

Author Response

1. We changed Figure 1 to Table 1 and modified numbering throughout the paper.

2. Table 1 (now Table 2) lists components that were ranked during the needs assessment interviews. The Component ‘Self-report questionnaire – SMS’ (Table 2 rank 6) was suggested by participants during the interviews as an important component for the MOvIT+™ intervention but was not found to be supported by any study included in the systematic review. To clarify this, we added this note under Table 2 : “The mention ‘none’ under ‘supporting literature’, means that this component emerged from the interviews, not from the systematic review”.

3. Line 545: We added a subtitle for the paragraph ‘Strengths and limitations’ to highlight that we list strengths and limitations in this paragraph.

Line 558: we added that the main limitation is time and resource commitment.

Line 565: we highlighted another limitation (no prioritization tool in the original Intervention Mapping Approach).

Line 571: we changed the title of this section to “5. Conclusions and future directions” to clarify. We moved one sentence referring to future directions in this section to regroup all future directions in a single section of the manuscript. 

Reviewer 3 Report

Thank you for the opportunity to review this article. This paper described the development of the Internet-based MOvIT+™  through an intervention mapping approach. In general, the paper is well organized, and the topic is relevant. For each section, it is clear and understandable. I like this paper and believe that it provides detailed information on the development of Internet-based interventions for future program developers.

Only 1 area needs revision:

-page 13, line 457. “Table 1 and grouped in two main categories of variables…” Please modify “Table 1” to “Table 2”.

Author Response

Page 13, now line 473. We modified “Table 1” to “Table 3” (because the numbering changed after Figure 1 was changed for Table 1).